# Isoprene-Emitting Tobacco Plants Are Less Affected by Moderate Water Deficit under Future Climate Change Scenario and Show Adjustments of Stress-Related Proteins in Actual Climate

**DOI:** 10.3390/plants12020333

**Published:** 2023-01-11

**Authors:** Susanna Pollastri, Violeta Velikova, Maurizio Castaldini, Silvia Fineschi, Andrea Ghirardo, Jenny Renaut, Jörg-Peter Schnitzler, Kjell Sergeant, Jana Barbro Winkler, Simone Zorzan, Francesco Loreto

**Affiliations:** 1Institute for Sustainable Plant Protection (IPSP), National Research Council of Italy (CNR), Via Madonna del Piano 10, 50019 Sesto Fiorentino, Florence, Italy; 2Institute of Plant Physiology and Genetics, Bulgarian Academy of Sciences, Acad. G. Bonchev Str., bl. 21, 1113 Sofia, Bulgaria; 3Institute of Biophysics and Biomedical Engineering, Bulgarian Academy of Sciences, Acad. G. Bonchev Str., bl. 21, 1113 Sofia, Bulgaria; 4Council for Agricultural Research and Economics, Research Center for Agriculture and Environment, Via di Lanciola 12/A, 50125 Cascine del Riccio, Florence, Italy; 5Institute of Heritage Science-CNR (ISPC), National Research Council of Italy (CNR), Via Madonna del Piano 10, 50019 Sesto Fiorentino, Florence, Italy; 6Research Unit Environmental Simulation (EUS), Helmholtz Zentrum München, Institute of Biochemical Plant Pathology, D-85764 Neuherberg, Germany; 7GreenTech Innovation Centre, Environmental Research and Innovation (ERIN) Department, Luxembourg Institute of Scienceand Technology (LIST), L-4362 Esch-sur-Alzette, Luxembourg; 8Department of Biology, University of Naples Federico II, Via Cinthia, 80126 Naples, Naples, Italy

**Keywords:** climate change, isoprene, photosynthesis, protection, water stress, proteomics

## Abstract

Isoprene-emitting plants are better protected against thermal and oxidative stresses, which is a desirable trait in a climate-changing (drier and warmer) world. Here we compared the ecophysiological performances of transgenic isoprene-emitting and wild-type non-emitting tobacco plants during water stress and after re-watering in actual environmental conditions (400 ppm of CO_2_ and 28 °C of average daily temperature) and in a future climate scenario (600 ppm of CO_2_ and 32 °C of average daily temperature). Furthermore, we intended to complement the present knowledge on the mechanisms involved in isoprene-induced resistance to water deficit stress by examining the proteome of transgenic isoprene-emitting and wild-type non-emitting tobacco plants during water stress and after re-watering in actual climate. Isoprene emitters maintained higher photosynthesis and electron transport rates under moderate stress in future climate conditions. However, physiological resistance to water stress in the isoprene-emitting plants was not as marked as expected in actual climate conditions, perhaps because the stress developed rapidly. In actual climate, isoprene emission capacity affected the tobacco proteomic profile, in particular by upregulating proteins associated with stress protection. Our results strengthen the hypothesis that isoprene biosynthesis is related to metabolic changes at the gene and protein levels involved in the activation of general stress defensive mechanisms of plants.

## 1. Introduction

Over a long evolutionary period, plants have developed different and complex mechanisms in response to stress factors. Under the present rapid global environmental changes, plants are even more prompted to quickly evolve adaptive and protective strategies. In the last decades, a large bulk of literature was produced with regard to plant defense mechanisms against abiotic stress, e.g., [1,2]. Among the protective mechanisms adopted by plants, it was suggested that the emission of volatile organic compounds (VOCs), and especially isoprene (the most abundant VOC emitted by plants), can play an important role in mitigating stress tolerance [3]. Moreover, plastidic isoprenoid biosynthesis through the methylerythritol 4-phosphate (MEP) pathway also generates non-volatile isoprenoids, such as carotenoids and xanthophylls, which are important in protecting the photochemistry of photosynthesis [4,5]. Isoprene itself is implicated in the protection of the photosynthetic apparatus by enhancing the integrity of thylakoid membranes [6,7,8], and by reducing stress-induced reactive oxygen and nitrogen species (ROS/RNS) [9,10,11,12]. Velikova et al. [8,13] demonstrated that the inhibition of isoprene biosynthesis in transgenic poplar modifies the lipid saturation and protein profile of chloroplast membranes, negatively affecting photosystem II photochemical efficiency and enhancing susceptibility to stress in non-emitting leaves. Isoprene-emitting leaves are characterized by the lower non-photochemical quenching of fluorescence under both control [8] and stressful [14,15,16] conditions. This finding was attributed to an improved elasticity and a sustained homogeneous distribution of photosystems in the photosynthetic membranes [16]. 

While the protective effect of isoprene on photosynthetic performance at high leaf temperatures seems to be proved beyond any doubt, there is still a debate as to whether isoprene may also protect against water deficit stress (hereafter simply called water stress) under actual or future climate conditions. Isoprene production is closely linked to photosynthesis, as an appreciable amount of carbon fixed by photosynthesis is shunted into the MEP-pathway [3,17,18]. When photosynthesis is depressed or completely inhibited by water stress-induced stomatal closure or by mesophyll, biochemical or photochemical limitations, isoprene emission still occurs at high rates [19,20,21]. This indicates that the contribution of additional (to photosynthesis) carbon sources is increased in stressed plants in order to sustain isoprene production [22,23]. The occurrence of a switch of carbon source from freshly assimilated C to “old” C sources in drought or salt-stressed leaves was demonstrated by labelling experiments [20,24,25]. Sustained isoprene emission therefore suggests an important role of this compound in avoiding the irreversible deterioration of the photosynthetic apparatus under water-limiting conditions. Numerous studies have focused on sustained isoprene emission under water stress [20,21,26,27,28]. Transgenic isoprene-emitting tobacco plants (Nicotiana tabacum) were utilized in water stress trials, and their performances were compared with wild-type non-emitting tobacco plants. Ryan et al. [29] demonstrated that photosynthesis was better protected in isoprene-emitting than in non-emitting tobacco plants during water limitation. However, non-emitting plants produced more biomass under water stress, implying that carbon deployed into isoprene biosynthesis incurred a yield penalty [30]. Tattini et al. [28] used the same transgenic tobacco, but their experiment aimed to compare physiological and biochemical traits of isoprene-emitting and non-emitting plants exposed to severe water stress and subsequent re-watering. During the re-watering phase, isoprene emitting plants showed higher photosynthesis than non-emitters. Non-volatile isoprenoids like xanthophylls and abscisic acid were also higher in isoprene-emitting tobacco in comparison with non-emitters. Results by Tattini et al. [28] suggest that the whole MEP pathway is up-regulated in water-stressed plants and sustains photosynthesis after recovery from stress, as also demonstrated in poplar by Vanzo et al. [31]. These papers also supported previous experiments [21], indicating that, in emitting plants, isoprene acts as a short-term protectant, whereas non-volatile isoprenoids and phenylpropanoids protect against severe, long-term damage. There is also growing evidence that isoprene biosynthesis primes the gene expression in unstressed plants, particularly up-regulating the genes of the phenylpropanoid pathway [12,32,33,34].

The future climate will likely be warmer, drier, and characterized by continuously rising CO_2_ in the atmosphere [35]. For example, the beneficial effect of isoprene may diminish under future climatic conditions, as increasing CO_2_ concentrations decrease the physiological and metabolic impact due to isoprene emission [36]. The combined impact of these changes on plants may be very different from the effect of single changing factors, e.g., the negative effects of water stress may be attenuated when stomata close in response to rising CO_2_ [37]. In the present work, we compared the ecophysiological performances of isoprene-emitting and non-emitting tobacco lines experiencing water stress (WS) and recovery in actual climate (28 °C and 400 ppm CO_2_) and future climate (32 °C and 600 ppm CO_2_) scenarios. Moreover, we characterized the proteomic profiles of WS plants in the actual climate scenario to further investigate possible changes related to the protective mechanisms of isoprene.

## 2. Results

### 2.1. Physiological Parameters

At the beginning of the experiment (T0), isoprene non-emitting plants (NE-WT) showed similar values for photosynthesis and ΦPSII, both under actual (AC) and future (FC) climate scenarios in well-watered condition (WW) (Figure 1a and Figure 2a). Isoprene-emitting plants (IE-H) displayed higher photosynthesis and ΦPSII in the FCWW than in the ACWW scenario (Figure 1a and Figure 2a). Stomatal conductance was similar among plants (NE-WT and IE-H) and scenarios (AC and FC) at T0 (Figure 1c). The non-photochemical quenching (NPQ) was similar among all plants and scenarios, and was comparably low at T0 (Figure 2c). Water stress reduced photosynthesis and stomatal conductance of NE-WT plants already at T1, whereas the photosynthesis of IE-H plants was less affected in actual climate (Figure 1b). The PSII operating efficiency and the NPQ of fluorescence were not yet affected by WS (Figure 2b,d), although ΦPSII started to decrease in all samples. At this time point, in well-watered and FC conditions, both NE-WT and IE-H plants showed a decrease in photosynthesis and ΦPSII compared to IE-H plants (Figure 1a and Figure 2a). When the WS became more severe (T2), photosynthesis (Figure 1b), stomatal conductance (Figure 1d), and ΦPSII (Figure 2b) of NE-WT plants under FCWS were strongly affected, whereas isoprene-emitting plants under ACWS and FCWS conditions showed similar photosynthesis, stomatal conductance and ΦPSII values such as NE-WT ACWS (Figure 1b,d and Figure 2b). The NPQ increased drastically in all FCWS plants at T2 but comparably less so in IE-H tobacco than in NE-WT (Figure 2d). When the stress became severe (T3), the photosynthesis (Figure 1b), stomatal conductance (Figure 1d) and ΦPSII (Figure 2b) of all plants were significantly reduced compared to irrigated controls (ACWW and FCWW), not only in the FCWS treatment (as for T2) but also in the ACWS treatment. The NPQ further increased in FCWS plants as well as in ACWS plants with respect to controls (Figure 2d). However, the NPQ of IE-H lines still remained lower than the NPQ of NE-WT tobacco, especially under the FCWS treatment. At this point, IE-H plants showed a higher photosynthesis and ΦPSII compared to NE-WT plants in the FCWW condition, as well as a higher relative water content (RWC%) (Appendix A). Finally, five days after ending the WS (T4), re-watered ACWS and FCWS plants showed similar or even higher (especially in NE-WT) photosynthesis (Figure 1b), stomatal conductance (Figure 1d) and ΦPSII (Figure 2b) than ACWW and FCWW control plants (Figure 1a,c and Figure 2a). Remarkably, all parameters were, in general, 50% lower than those measured at the beginning of the experiment, both in controls and stressed plants, implying a strong ageing effect on leaf physiology across the time course of the experiment, and independent of the stress. The NPQ dropped again to very low values in all samples at T4, except for FCWW (control) plants, where NPQ showed a sudden and unexpected rise (Figure 2c,d). In the absence of a plausible explanation directly associated with our experiment, we think that the NPQ rise might have been caused by other reasons, as discussed below. 

### 2.2. Genes Expression

The dehydrin gene, NtERD10A, which is an indicator for drought stress, showed an increase in gene expression after WS treatment, particularly in the IE-H line (Figure 3a). The expression level of the isoprene synthase gene (ISPS), which is responsible for isoprene biosynthesis, was measurable only in the IE-H line (Figure 3b). After WS, the ISPS gene expression decreased more than three times compared to controls, returning to control levels after recovery.

### 2.3. Proteins

A total of 359 different protein spots were identified with proteomic analysis. The complete list is reported in Appendix A. Biological processes and functions were identified for most of the proteins (http://www.uniprot.org/, accessed on November 2016) (Figure 4). According to our attribution to biological processes, the proteins that were differently abundant belonged mainly to the photosynthesis and ATP synthesis and stress response categories (Figure 4). Fifty-five proteins significantly varied between genotypes and/or because of stress occurrence (Appendix A). 

In particular, in control conditions, three proteins were differently abundant in IE-H compared to NE-WT, i.e., associated with isoprene biosynthesis. The three proteins which were differently abundant in the IE-H line compared to NE-WT in control conditions (ACWW at T0) were an atpB gene product (spot 732) involved in light reactions of photosynthesis/ATP synthesis (Figure 5a), the proteasome subunit alpha type-6 (spot 1458), a protease related to ubiquitin-dependent protein catabolic process (Figure 5b), and the putative glutathione S-transferase (spot 1553) related to the oxidation/reduction process (Figure 5c). Figure 6a–e show proteins that are statistically different between IE and NE under WS (T3) and actual climate (ACWS) conditions and were more abundant compared to controls (ACWW). These were generally proteins attributed to stress as biological process and function (Figure 4), and were more abundant in IE-H than in NE-WT.

Proteins more abundant in IE-H than in NE-WT at T3 included the stress protein DDR48-like (spot 822) (Figure 6a), its isoform 823 spot (Figure 6b), and the abscisic stress-ripening protein 2 (1926 spot) (Figure 6e). The 17.6 kDa class I heat shock protein 3-like (spot 1758) was 15 and 5 times higher following ACWS in IE-H and NE-WT, respectively, compared to control conditions (Figure 6c), and the 17.8 kDa class I heat shock protein-like (spot 1763) was 10 and 3.5 times higher following ACWS in IE-H and NE-WT, respectively (Figure 6d). The exception was Rubisco activase 2 (spot 970), a protein involved in photosynthesis and found to be more abundant in NE-WT than in IE-H plants following ACWS (Figure 6f).

## 3. Discussion

Isoprene emission helps plants to cope with abiotic stresses [3], but is this becoming an advantage under predicted future climate conditions with prolonged periods of drought, high temperature and climate extremes combining both stress factors? To respond to this question we performed an experiment in which tobacco plants that were already characterized for their isoprene-driven protection against water stress (WS) [38], were exposed to actual and future climate change scenarios, in both unstressed and water-stressed conditions. We also tried to introduce new evidence of possible mechanisms explaining isoprene-driven protection based on a proteomic analysis. 

Our study highlighted that in a future climate (FC) change scenario characterized by elevated CO_2_ concentration and air temperature, all plants showed a more rapid decrease of photosynthesis and photochemistry parameters in WS conditions (Figure 1b). During the experiment in the FC scenario, the time points at which isoprene-emitters showed a better performance of photosynthesis when compared to non-emitting plants was under a 50% reduction of water supply (T2) (Figure 1b). Interestingly, T2 under FCWS was also the stage at which isoprene-emitters showed a much lower increase of the NPQ compared to that observed in non-emitting wild types (Figure 2d). This confirms that isoprene emission capacity is often associated with NPQ reduction, which was explained by a higher stability and elasticity of photosynthetic membranes, especially when challenged by moderate stresses [14,16]. 

We also noted that all plants only partially recovered their original photosynthesis parameters when re-watered, perhaps due to the fact that tobacco plants aged rapidly. Indeed, we expected a complete recovery, as a long recovery time was allowed. Remarkably, most photosynthetic parameters of WS plants were significantly higher than in well-watered (WW) plants at T4 (after recovery), and again independently of isoprene emission or climate scenario. The unexpected rapid decrease in photosynthesis in all plant groups might have lessened the effect of isoprene. It has been surmised that the protective effect of isoprene is limited to moderate stress [21] or to a morning/daytime course [38], while it is complemented by other, more efficient antioxidants when the stress becomes heavier and more prolonged. The large reduction of photosynthetic properties in WW plants and the unrecoverable effect of WS, here interpreted as indications of fast leaf ageing, might have reduced isoprene emission both at ambient and higher CO_2_ concentrations [39]. Unfortunately, we could not measure isoprene emission with our experimental set-up to confirm this hypothesis. Low isoprene emission, if reflecting a reduced synthesis and not an increased resistance due to stomatal closure often observed under WS [40], may not have been able to protect leaves, as the anti-stress property of isoprene is often concentration-dependent [41]. 

All plants exposed to the FCWW scenario surprisingly showed a very high NPQ at the end of the experiment (T4). We associate such a large increase of NPQ to the fast reduction of photosynthesis, perhaps indicating leaf ageing, which was particularly strong in well-watered plants. Perhaps plants exposed to the enhanced CO_2_ concentration dissipate less electron transport by photorespiration and must therefore activate non-radiative mechanisms of energy dissipation to cope with light energy pressure without generating damaging oxidative species. Notably, however, photosynthesis dropped significantly less and NPQ increased significantly less in IE-H than in NE-WT plants, confirming the general effect of isoprene as a photosynthesis stabilizer and a quencher of non-photochemical dissipation routes [15]. Perhaps this also indicates a delayed ageing of isoprene-emitting leaves if isoprene proxies’ cytokinin presence, as suggested by Dani et al. [42]. More recent studies, however, indicated that leaf senescence is hastened in isoprene-emitting leaves [43]. As isoprene synthesis is expected to drop with rising CO_2_ [3], so perhaps low concentrations of isoprene indeed support extended leaf life. 

Previous reports have shown that isoprene emission in poplar reshapes the leaf proteome [34,43], in particular the chloroplast proteome [13]. In tobacco, the effect of isoprene on the overall proteome was much less pronounced. One option is that isoprene impacts the proteome only under (severe?) stress conditions in tobacco plants. The protective effect of isoprene on photosynthesis in WS plants has often been observed, e.g., [20,21,38]. The less pronounced protein changes appear to agree with the reduced protective effect of isoprene on photosynthesis seen in our experiments. However, we could detect differences that link protein expression with isoprene production, particularly under water deficit conditions. 

Given the function of isoprene, changes in the abundance of proteins related to photosynthesis and to stress response were of particular interest. Specifically, we speculate that the capacity to emit isoprene: (i) might have reduced the impact of WS on the proteome; and (ii) might have changed the proteomic profile, particularly contributing to the activation of specific pathways that help isoprene emitting plants to cope with stress under future climate conditions. This second effect, in particular, supports a much wider regulatory role of isoprene, as also indicated in recent works [34,44,45]. 

We were particularly interested in changes involving proteins of the chloroplast, where isoprene is synthesized and thought to exert its function [14,16], and where proteins that mirror stress-induced activation of defensive secondary metabolites are also present [13]. The following discussion will be limited to selected proteins localized in chloroplasts or those that are stress-associated. Three proteins were found to be significantly higher in well-watered isoprene-emitting tobacco plants (Figure 4b), and are therefore most likely representative of processes that are related to isoprene emission, independent of stress occurrence. 

The atpB gene product (732) was significantly higher in IE-H than in NE-WT control and WS plants (Appendix A). We identified a total of six different isoforms of this protein (Appendix A, spots 732, 757, 764, 767, 769, 854). The atpB gene is encoding the α-subunit of chloroplast ATP synthase. Chloroplast ATP synthase catalyzes the light-driven synthesis of ATP and acts as a key feedback regulatory component of photosynthesis. Rott and colleagues [46] reported that a low level of the chloroplastic ATP synthase represses photosynthesis because it activates photoprotective mechanisms, such as NPQ, and downregulates linear electron flux. The increased amount of this protein in the IE-H tobacco plants, on the other hand, might be associated with the maintenance of low NPQ in isoprene-emitting plants, both in physiological conditions [15] and during stress [16,47]. 

The abundance of the proteasome subunit alpha type-6 is almost double in IE-H compared to NE-WT plants at T0 (Figure 5b). This protein belongs to a bigger family group forming the 26S proteasome system that together with ubiquitin controls the turnover rates of misfolded and damaged proteins, as well as numerous regulatory proteins (UPS) [48,49,50]. Many studies also reported that proteasome abundance and activity may increase during plant development and in response to environmental stress conditions [48]. It may be speculated that an increased abundance of the proteasome subunit alpha type-6 (1458, Appendix A) protein in IE-H plants is a result of the signaling effect of the isoprene activating synthesis of several secondary metabolites for their rapid involvement as anti-stress compounds [33,33,34,51,52,53]. In particular, the proteasome subunit alpha type-6 (1458) may lead to the enhanced responsiveness of signal transduction pathways and increased stress resistance by accelerating the removal of damaged proteins [54]. 

The third protein that we found to be more abundant in the IE-H line was a putative glutathione S-transferase (1553, Appendix A). Glutathione S-transferases (GSTs) are important in maintaining cellular redox homeostasis and protecting plants against oxidative damage (Figure 5c). Arabidopsis plants overexpressing GST were tolerant to stress induced by phenol [55] and to salinity and oxidative stresses [56]. Transgenic tobacco plants overexpressing Gst-cr1, a gene encoding a GST from cotton, showed enhanced resistance to oxidative damage [57]. Roxas et al. [58] report that the overexpression of Nt107, a tobacco GST with glutathione peroxidase activity (GST/GPX), induces tolerance to several stresses such as low and high temperature, salt, and herbicide exposure. The involvement of isoprene in antioxidant processes has been debated for some time. While isoprene was initially discussed as an antioxidant [9,11], it has recently been proposed that isoprene indirectly influences antioxidants, thereby contributing to reduce ROS levels and to regulate the redox state of the cells. Isoprene may prime several antioxidant systems, including the classic enzymatic antioxidants of the Halliwell-Asada cycle. A higher capacity to stimulate other antioxidant defenses in isoprene-emitting plants, including xanthophylls and phenylpropanoids, was earlier reported [28], although other potential antioxidants, such as α-tocopherol [51], or signaling molecules such as jasmonic acid [34], may be depleted when isoprene synthesis is active. Monson et al. [59] recently suggested that isoprene may play a central role in the growth-defense trade-off, providing a signal to shift resources to a defensive metabolism when prompted by stresses. 

The impact of WS on selected tobacco proteins was further examined by calculating the ratio between the proteins in stressed plants and in controls. Among chloroplast proteins related to photosynthesis, we generally found upon WS a more stable proteome in IE-H plants than in NE-WT plants. However, Rubisco activase large isoform (RCA2) was significantly less abundant in IE-H than in NE-WT plants following the stress (Figure 6f). Rubisco activase is an important enzyme that, through carbamylation of the ribulose-1,5-bisphosphate carboxylase/oxygenase (Rubisco) active site, regulates its activity and is itself regulated by the redox state of the chloroplast [60]. The large isoform RCA2 is present both in the chloroplast stroma and in thylakoids, and is most responsive to stresses [61]. The thermolability of RCAs may cause the inhibition of photosynthesis during moderate heat stress [62,63], but proteomic analyses have also shown that the abundance of RCAs changes when plants are exposed to other abiotic stress treatments [64,65,66]. In particular, the RCA2 content significantly increased to protect other functional proteins from damage under stress conditions [61]. Our results suggest that plants emitting isoprene exhibit lower stress levels and also avoid the accumulation of RCAs, which can imply less damage to the photosynthetic apparatus in the presence of isoprene [16]. Contrary to RCA2, several proteins increased significantly more in WS IE-H than in NE-WT tobacco plants, and some of them were still upregulated during the recovery phase (T4), suggesting that the continued accumulation of these proteins might increase WS resistance in isoprene-emitting plants (Figure 6). Three of these proteins were small heat shock proteins (sHsps spots 1758, 1763, 1777, Appendix A) which are also induced by various abiotic and oxidative stresses [67]. sHsps belong to the chaperone system that prevents heat stress-induced denatured proteins from forming non-specific aggregates that severely impede normal cellular functions [68,69]. The increasing expression of heat shock proteins in concert with high ISPS protein levels have also been reported under water limitation in date palm [70]. sHsps have also been shown to be amphitropic and to increase the molecular order of the lipid bilayer, conferring thermotolerance [71,72,73]. It is interesting that a similar role was proposed for isoprene [7,16,74]. We interpret this finding as another piece of evidence of priming induced by the presence of isoprene [53]. Proteins primed by isoprene may assist in protecting membrane functionality, performing a function previously only attributed to isoprene. Some of the accumulating sHsps (e.g., Hsp17.6CII) may also increase catalase activity in the peroxisome, thus contributing to an improved stress resistance indirectly via the stimulating of enzymatic antioxidants [75]. Finally, higher levels of Hsp17.8 together with the protein AKR2A enhance the targeting efficiency of chloroplast membrane proteins, as demonstrated by Kim et al. [76]. This is in line with the main finding that isoprene absence in transgenic poplar triggers a re-arrangement of the chloroplast protein profile to minimize the negative stress effects [13], and in general by a more efficient array of protections against stresses [44]. The DNA damage-responsive protein DDR48 (spots 822, 823, Appendix A) was also more abundant in isoprene-emitting tobacco plants than in non-emitters after imposing the WS. This protein is associated with increased gene transcription in response to treatments producing DNA lesions or heat-shock stress in organisms other than plants. However, it has not been investigated in plants yet [77].The abscisic stress-ripening protein 2 (ASR2) was twofold higher in the IE-H WS tobacco plants with respect to NE-WT. This protein possesses a chaperone-like and transcription factor activity [78] which can be induced by abscisic acid (ABA) and abiotic stresses, primarily salinity and drought [79]. The overexpression of ASR genes resulted in the increased tolerance of transgenic Arabidopsis and tobacco to water/osmotic stress [80,81], suggesting that our IE-H line displays similar tolerance. Tattini et al. [28] reported that ABA was enhanced in drought conditions in transgenic isoprene-emitting tobacco. Moreover, a direct relationship between isoprene emission and foliar ABA content was reported earlier [82]. Isoprene may proxy ABA content or the introduction of ISPS, and hence the ability to emit isoprene may have strengthened the carbon flow through the MEP pathway, which also may lead to increased ABA biosynthesis via xanthoxin catabolism. However, there are also cases in which a trade-off between isoprene and non-volatile MEP-derived isoprenoids is present, for example with carotenoids [20]. Moreover, the physiological effects of high ABA (e.g., stomatal closure) have not been associated with high isoprene emissions, but stomata do open when isoprene biosynthesis is inhibited by fosmidomycin and ABA is potentially reduced [82]. The relationship between ABA and other MEP-derived phytohormones is clearly not yet fully elucidated, but ASR2 behavior may represent another evidence that isoprene emission is generally associated with higher levels of ABA, which has positive feedback on water stress resistance [83,84]. 

## 4. Materials and Methods

### 4.1. Plant Materials, Growth and Sampling Conditions

Wild-type plants that do not emit isoprene (NE-WT) and isoprene-emitting plants of a transgenic line (12 H) homozygous for isoprene synthase (isoprene emitting, IE-H) of Nicotiana tabacum (cv Samsun) were used for these experiments [11]. The first experiment, dedicated to gas-exchange measurements, was performed using the environmental simulation facilities at the Helmholtz Munich, Germany, and the second, also involving gene expression and proteomic measurements, was carried out at CNR in Florence, Italy.

#### 4.1.1. First Experiment

Plants were grown from seeds under two different CO_2_ concentrations and temperatures in two greenhouse cabins using plastic pots (2,2 L) filled with commercial soil. One group was grown at 400 ppmv atmospheric CO_2_ and a day/night temperature of 28 °C/24 °C (actual climate scenario, AC); the other group was grown at 600 ppmv atmospheric CO_2_ and a day/night temperature of 32 °C/28 °C (future climate scenario, FC, i.e., with the concurrent increase of temperatures and CO_2_ concentrations). Both groups of tobacco plants were exposed to the same photoperiod (16/8 h of light/dark) and light intensity (200–240 μmol m^−2^ s^−1^ of photosynthetic photon flux density (PPFD) at the canopy level). Light was provided at the steady state level with high-pressure sodium vapor lamps (Philips Son-T agro, Amsterdam, Netherlands). Plants were fertilized weekly with an NPK commercial solution. After 38 days, plants were further divided into four subgroups, each made by 48 individuals, and each subgroup was transferred into a different highly controlled phytotron chamber (for details see [31,85]). All plants were cultivated at 500 µmol m^−2^ s^−1^ PPFD at canopy level and were subjected to the following treatments in the four chambers: (1) AC plants continued to grow under the actual climate scenario and were well-watered (ACWW); (2) AC plants were exposed to g followed by a re-watering phase (ACWS); (3) FC plants continued to grow under the future climate scenario and were well-watered (FCWW); and (4) FC plants were exposed to WS and re-watered (FCWS). Inside each chamber were four gas-tight sub-chambers (made of UV-transparent acrylic glass (∼1 m^3^), allowing for the separation of the isoprene-emitting plants from the non-emitters). Air, in controlled conditions (temperature, humidity and [CO_2_]), was flushed inside each sub-chamber, inside which air temperature and relative humidity sensors were also present. After five days of acclimation in the phytotron chamber, ecophysiological parameters were measured as described below when all plants received the same amount of water (240 mL day-1–100% of water supply) (T0). All measurements were performed on the first fully expanded leaf, which was the third to fourth leaf from the apical meristem. The water stress was then applied to both treatments, ACWS and FCWS, by progressive reduction of the supplied water, and ecophysiological measurements were repeated in all four treatments at the following time points: after two days with 70% of water supply in ACWS and FCWS (T1); after four more days with 50% of water supply in ACWS and FCWS (T2); and after two more days without watering ACWS and FCWS plants at all (T3). The WS plants were again irrigated in the same manner as the control plants, and a further round of measurements was carried out after five days of recovery, when the soil water content of WS plants was again close to controls (T4). Five isoprene-emitting and five non-emitting plants of each of the four treatments were harvested at the end of the stress phase (T3), and five more plants of each genotype and treatment were harvested after stress recovery (T4). The experimental setup is graphically depicted in Appendix A, where the developmental stages of the plants are shown. 

#### 4.1.2. Second Experiment

A separate experiment was repeated in the same condition of the actual climate scenario (AC) as described above, and frozen samples at T0, T3 and T4 were collected and used for gene expression and proteomic analyses, as shown below.

### 4.2. Gas Exchange Measurements and Chlorophyll Fluorescence

A Licor 6400 XT portable photosynthesis system (LI-COR, Lincoln, NE, USA) was used to measure photosynthesis (Pn) and stomatal conductance (gs). Measurements were performed by placing a portion of a leaf inside the 2 cm^2^ LI-COR leaf cuvette where the leaf was exposed to the following conditions: 400 (ACWW or ACWS) or 600 (FCWW or FCWS) ppmv of CO_2_, and leaf temperatures of 28 °C (ACWW or ACWS) or 32 °C (FCWW or FCWS). Light intensity, air flow rate, and relative humidity were set at 1000 μmol m^−2^ s^−1^ PPFD, 0.3 L min^−1^ and 40–50%, respectively, in all sampled leaves. Measurements were recorded when a steady-state gas exchange was reached. Fluorescence parameters were measured using a MINI-PAM Photosynthesis Yield Analyzer (Heinz-Walz, Effeltrich, Germany). The minimal (F0) and maximal (Fm) chlorophyll fluorescence was measured in the dark after 30 min of adaptation; Fm’, the maximal fluorescence, and Fs, the steady-state fluorescence, were measured in leaves at the growth light conditions (500 μmol m^−2^ s^−1^ PPFD) and used to calculate the PSII operating efficiency (ΦPSII = Fm’ − Fs/Fm’) and the non-photochemical quenching of fluorescence (NPQ = (Fm − Fm’)/Fm’) [86,87].

### 4.3. Relative Water Content, Fresh and Dry Weight Measurements

The relative water content (RWC) was determined in fully expanded leaves collected from five different plants per treatment and per time of measurement, according to [88]. Fresh weight was recorded immediately after harvesting and dry weight after placing the plant material in an oven at 65 °C for five days. The turgid weight was measured after leaving the leaves submerged in distilled water for the whole night.

### 4.4. Gene Expression Analysis

Total RNA was extracted from T0, T3 and T4 leaf samples using the RNeasy Plant Mini Kit (Qiagen, Hilden, Germany), genomic DNA was removed using DNAase I (Promega, Madison, WI, USA), and the concentration and purity of the RNA samples were determined using BioPhotomether (Eppendorf, Hamburg, Germany). RNA samples were reverse-transcribed using the GoScript Reverse Transcription System (Promega, Madison, WI, USA). The expression of the isoprene synthase (ISPS) gene was determined by means of real-time PCR, using primers designed on the Populus alba ISPS sequence (Genbank accession number AB198180.1). The Nicotiana tabacum genes L25 (Genbank accession number L18908) and NtERD10A (Nicotiana tabacum Early Responsive to Dehydration 10A) (Genbank accession number AB049335.1) were used as reference gene and water stress indicators, respectively. All primers were designed using the software Primer3 v 0.4.0 (http://bioinfo.ut.ee/primer3-0.4.0/primer3/, accessed on November 2016) following the manufacturer’s guidelines. The primers used for each gene were: L25 forward: 5′ CCCCTCACCACAGAGTCTGC 3′; L25 reverse: 5′ AAGGGTGTTGTTGTCCTCAATCTT 3′; NtERD10A forward: 5′ TTGCTGAGTTCTGAAGCGTG 3′; NtERD10A reverse: 5′ ACGAGGCACATGATACAACG 3′; ISPS forward: 5′ CTGTTTGGAGCATTGAAGCA 3′; ISPS reverse: 5′ CCTCCACCACCTTGATGTCT 3′. Real time PCR was performed using Power SYBR Green Mastermix (Applied Biosystems, Foster City, CA, USA) on the Applied Biosystems Step-One Real-Time PCR System (Applied Biosystems). The reaction conditions consisted of 2 μL of cDNA and 0.2 μM primers in a final volume of 15 μL. The cycling conditions were as follows: 95 °C for 5 min, followed by 40 cycles of 95 °C for 15 s, and at 60 °C for 1 min. Appropriate no-RT and non-template controls were included in each 96-well PCR reaction. Reactions were run in triplicate in three independent experiments. Expression data were normalized to the geometric mean of housekeeping gene L25 to control the variability in expression levels and were analyzed using the 2 -ΔΔCT method described by Livak and Schmittgen [89].

### 4.5. Proteins Extraction and Analysis

Leaf samples for protein extraction were harvested at the time points T3 and T4. Proteins were solubilized with 200 µL of 2D-DIGE labelling buffer (7 M urea, 2 M thiourea, 2% (*w*/*v*) CHAPS, 30 mM Tris, containing protease inhibitors mix from GE Healthcare). Protein extracts were clarified by centrifugation at 10,000× *g* for 10 min at 4 °C. The determination of protein concentrations, labelling and 2D-DIGE were then carried out as described in [90]. Digestion was carried out using a Freedom EVO II workstation (Tecan, Männedorf, CH) as described in [91]. MS and MS/MS spectra were submitted for NCBInr database-dependent identification against the NCBI database limited to the taxonomy Viridiplantae (2,471,722 sequences) (http://www.ncbi.nlm.nih.gov from January till June 2016) on an in-house MASCOT server (Matrix Science, www.matrixscience.com accessed on November 2016). A second search was carried out against an EST (expressed sequence tag) Nicotiana database containing 2,582,928 sequences. The parameters used for these searches were a mass tolerance of MS 100 ppm, a mass tolerance of MS/MS 0.5 Da, fixed modifications of cysteine carbamidomethylation, variable modifications of methionine oxidation, and the double oxidation of tryptophan and of tryptophan to kynurenine. Proteins were considered as identified when at least two peptides passed the MASCOT-calculated 0.05 threshold scores (a score of 52 for all NCBI Viridiplantae queries and 50 for the EST queries, respectively).

### 4.6. Statistical Analysis

Ecophysiological and gene expression data are shown as means (*n* = 5) ± standard errors (SEs). The normality of data distribution was tested using the Shapiro–Wilk Normality Test. Significant differences within genotypes at different time points and conditions were analyzed using a one-way analysis of variance (ANOVA) (* *p* value <0.05; ** *p* value < 0.01). SigmaPlot was used for all of these analyses (Systat Software Inc., San Jose, CA, USA).

Protein spots with an absolute ratio of at least 1.5-fold and *p*-values < 0.05 were considered as differentially expressed with statistical significance. On proteomics data, an ANOVA was used for homoscedastic data, while the Welch Test was chosen if heteroscedasticity was detected. Post hoc tests used were either the LSD Fisher or the Duncan Waller. Proteomic data are shown as means (*n* = 3) ± standard errors (SEs). Data were processed with custom R (v. 3.2.1) procedures. 

## 5. Conclusions

In summary, our experiments generally show improved physiological resistance to moderate water stress under future climate conditions in tobacco plants engineered to produce and emit isoprene. The positive effect of isoprene on photosynthesis and on photosynthetic electron transport was visible and significant when the stress was still moderate. Future climate conditions (higher temperatures and enhanced atmospheric CO_2_ concentration) may further aggravate WS impacts on photosynthesis, which isoprene may again be able to protect from, albeit only in the early stages of the stress. Despite scarce physiological evidence of a positive action of isoprene in our short-term experiment, making plants able to emit isoprene also affected several proteins, mostly inducing the accumulation of those proteins associated with the stress protection metabolism. Our results strengthen the idea that isoprene, besides having a specific role in protecting photosynthetic membranes, may also induce or elicit metabolic changes at the gene and protein levels involved in activating stress defensive mechanisms. These mechanisms might prove themselves to be more useful when the stress is recurrent over the plant’s life. 

## Figures and Tables

**Figure 1 plants-12-00333-f001:**
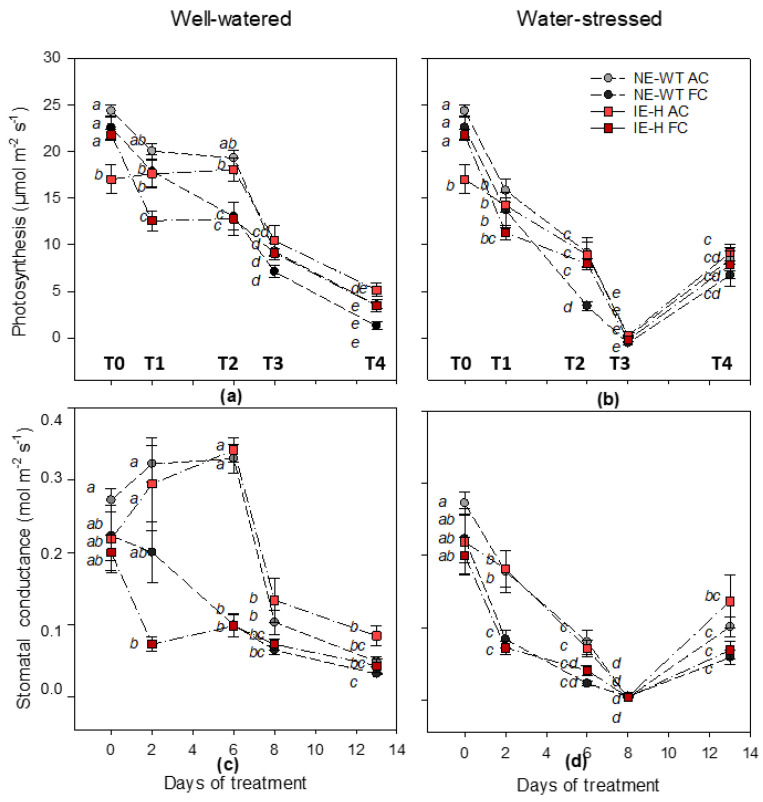
Photosynthesis and stomatal conductance time courses. Photosynthesis (**a**,**b**) and stomatal conductance (**c**,**d**) of non-emitting wild-type = NE-WT (triangles; circles) and isoprene-emitting line = IE-H (diamonds; squares) in actual climate scenario (AC: 400 ppmv and 28 °C; light grey = NE-WT and light red = IE-H) and future climate scenario (FC: 600 ppmv and 32 °C; dark grey = NE-WT and dark red = IE-H) in well-watered WW (**a**,**c**) and water-stress WS (**b**,**d**) conditions. Measurements were carried out before imposing the stress at day 0 (=T0), after 2, 6 and 8 days of water stress (WS; T1, T2 and T3, respectively), and after 5 days of recovery by re-watered plants (day 13 = T4). Control plants in well-watered conditions (WW) were also measured along the same time course. Means (*n* = 5) + SE are shown for each data point. For each treatment (WW, WS) a one-way ANOVA followed by a Tukey’s test was performed to define the statistical significance (*p* < 0.05) of mean differences along the time course of the experiment between genotypes (NE-WT, IE-H) in different climate conditions (AC, FC). Different letters show statistically significant differences.

**Figure 2 plants-12-00333-f002:**
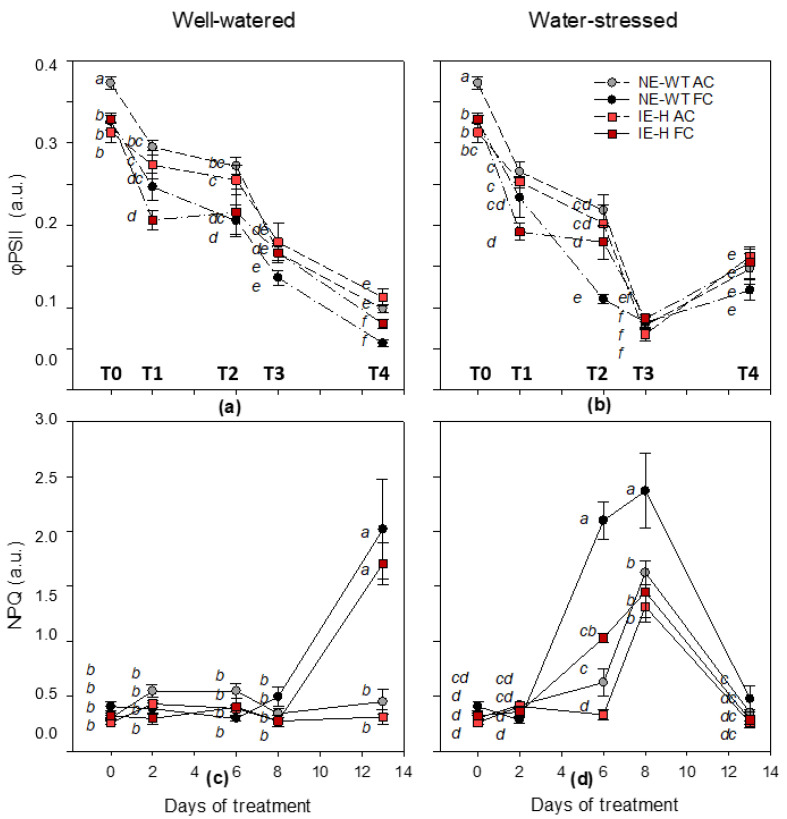
ΦPSII and NPQ time courses. Photochemical efficiency of PSII (ΦPSII; A, B) and non-photochemical quenching of fluorescence (NPQ; C, D) of non-emitting wild-type = NE-WT (triangles; circles) and isoprene-emitting line = IE-H (diamonds; squares) in actual climate scenario (AC: 400 ppmv and 28 °C; light grey = NE-WT and light red = IE-H) and future climate scenario (FC: 600 ppmv and 32 °C; dark grey = NE-WT and dark red = IE-H) in well-watered WW (**a**,**c**) and water-stress WS (**b**,**d**) conditions. Measurements were carried out before imposing the stress at day 0 (=T0), after 2, 6 and 8 days of water stress (WS; T1, T2 and T3, respectively) and after 5 days of recovery by re-watered plants (day 13 = T4). Control plants in well-watered conditions (WW) were also measured along the same time course. Means (*n* = 5) + SE are shown for each data point. For each treatment (WW, WS), a one-way ANOVA followed by a Tukey’s test was performed to define the statistical significance (*p* < 0.05) of mean differences along the time course of the experiment between genotypes (NE-WT, IE-H) in different climate conditions (AC, FC). Different letters show statistically significant differences.

**Figure 3 plants-12-00333-f003:**
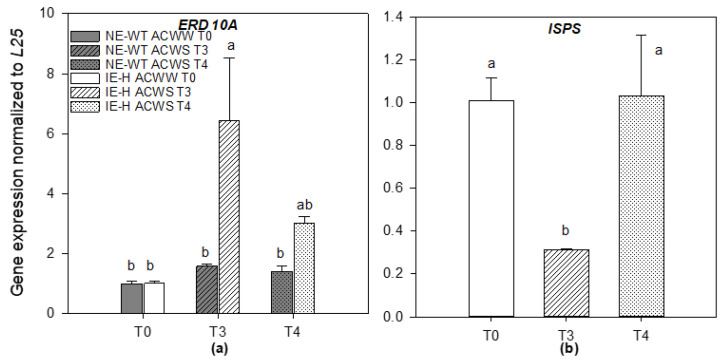
Expression of NtERD10A and ISPS genes in actual climate conditions. (**a**). Expression of the dehydrin NtERD10A gene was measured in non-emitting wild-type = NE-WT (grey) and isoprene-emitting line = IE-H (white) at different time points (T0: full color; T3: striped; T4: dotted). Levels from NtERD10A were normalized against L25 in each treatment. (**b**). Expression of *ISPS* gene was measured in IE-H line in each treatment. Levels from *ISPS* were normalized against L25 in each treatment. Means (*n* = 3) ± SE are shown. A one-way ANOVA followed by a Tukey’s test was performed to define the statistical significance (*p* < 0.05) of differences among means across the treatments. Different letters show statistically significant differences.

**Figure 4 plants-12-00333-f004:**
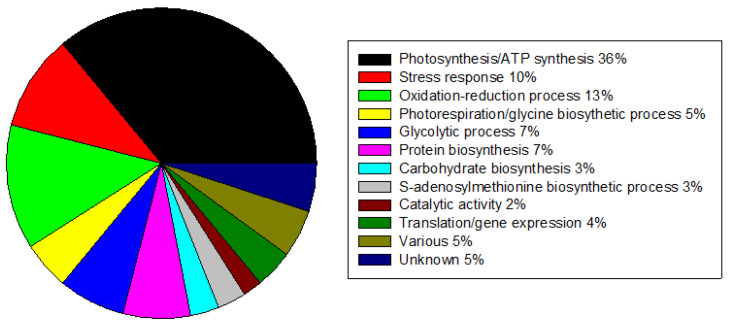
Proteomic overview in actual climate conditions. Proteins retrieved by our measurements of the proteome and separated into 12 different biological function groups. Groups were ordered accordingly to their percentage, calculated on a total of 359 different protein spots.

**Figure 5 plants-12-00333-f005:**
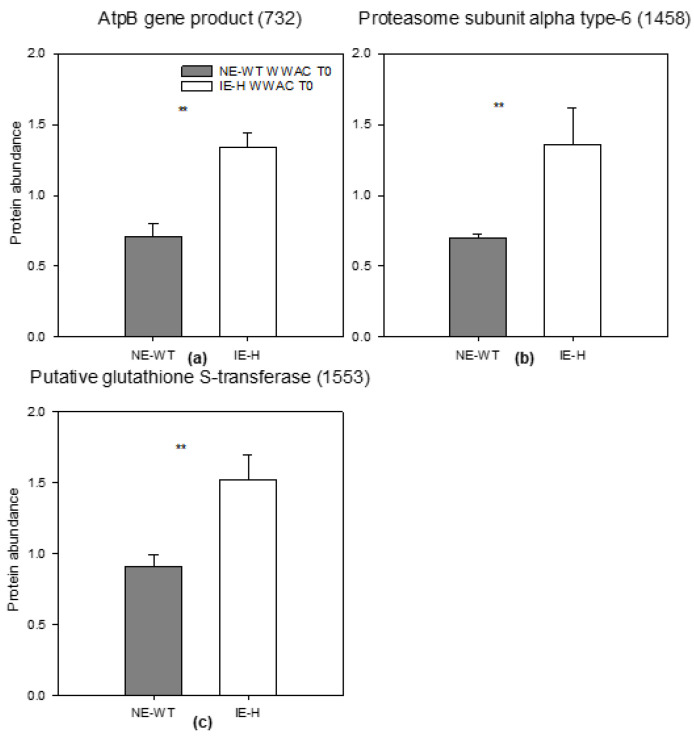
Different abundance of proteins in control conditions in actual climate. The abundances of atpB gene product (732) (**a**), proteasome subunit alpha type 6 (1458) (**b**) and putative glutathione S-transferase (1553) (**c**) in non-emitting wild-type = NE-WT (grey) and isoprene-emitting line = IE-H (white) under control condition (WW) are reported in control plants (at T0). Means (*n* = 3) + SE are shown. A Student’s *t*-test was performed to determine the statistical significance (** *p* < 0.01) of differences in the abundance of proteins between genotypes.

**Figure 6 plants-12-00333-f006:**
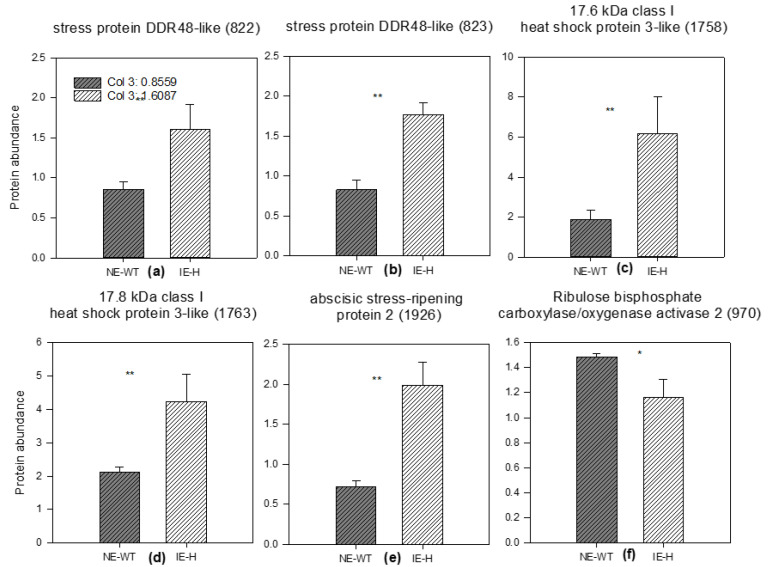
Different abundance of proteins in water-stressed conditions in actual climate. The abundances of stress protein DDR48-like (822, 823) (**a**,**b**), 17.6 kDa class I heat shock protein 3-like (1758) (**c**), 17.8 kDa class I heat shock protein-like (1763) (**d**), abscisic stress-ripening protein 2 (1926) (**e**) and Rubisco activase 2 (970) (**f**) are reported in severely water-stressed plants (at T3). Grey-striped bars show non-emitting wild-type = NE-WT, and white-striped bars are isoprene-emitting line = IE-H. Means (*n* = 3) + SE are shown. A Student’s *t*-test was performed to define the statistical significance (* *p* < 0.05; ** *p* < 0.01) of differences in the abundance of proteins between genotypes.

## Data Availability

Not applicable.

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
