# Peer review of "Isoprene-Emitting Tobacco Plants Are Less Affected by Moderate Water Deficit under Future Climate Change Scenario and Show Adjustments of Stress-Related Proteins in Actual Climate"

_plants, 2023, doi:10.3390/plants12020333_

Round 1

Reviewer 1 Report

I have no comments on the manuscript. The material is presented logically and fairly reliably.

Author Response

We would like to thank the reviewer for her/his supportive comment.

Reviewer 2 Report

The article is interesting and valuable. It could be published after minor technical corrections. 

Author Response

(The authors gave the same response as above.)

Reviewer 3 Report

The authors compared the changes in photosynthesis and proteomic properties between isoprene emitting and non-isoprene emitting tobacco plants during water stress in actual climate and future climate conditions. Their results strengthen that the isoprene biosynthesis is also related to the stress-related protein accumulations. The manuscript is overall well-written. The figure presentations would need improvement and some additional information on the experiments should be provided in the revised version.

1.      It is difficult to distinguish between the AC and FC treatments on Figures 1 and 2 because similar colors (gray vs. black, or orange vs red) are used. Please change their symbols

2.      Please provide information on the pot size, nutrient conditions and plant size in the materials and methods section, because effects of water stress on the plants frequently depend on those factors.

3.      Line 440: The authors used gas-tight sub chambers to separate the genotypes. However, I am concerned that the gas-tight chambers significantly reduce the CO2 concentrations in the chambers due to CO2 uptake by photosynthesis and also increase the air-temperature inside, perturbating the growth conditions. More detail information on the sub-chambers and how to deal with this issue will be needed in the methods section.  

4.      Line 314: “ “-subunit. The letter seems to disappear.

5.      Line 477-: The data regarding RWC appear not to be described and the authors need to mention the results and discussion.

Author Response

We would like to thank the reviewer for her/his supportive comment. Please also find below a point-to-point response to the reviewers’ comments. 

1. We have changed the symbols in figure 1 and 2, as suggested.

2. In the “Plant materials, growth and sampling conditions” section of Materials and Methods, we have added the following information in order to respond to the reviewer’s request, and also coherently modified Figure S1.

Line 428-429: Plants were grown from seeds under two different CO2 concentrations and temperatures in two greenhouse cabins using plastic pots (2,2 L) filled with commercial soil.

Line 437: Plants were fertilised weekly with a NPK commercial solution.

Line 463-464: The experimental setup is graphically depicted in Fig.S1, where also the developmental stages of the plants are shown.

3. In the “Plant materials, growth and sampling conditions” section of Materials and Methods, we have added the following information in order to respond to the reviewer request.

Line 447- 449: Air, in controlled conditions (temperature, humidity and [CO2]), was flushed inside each sub-chamber, inside which air temperature and relative humidity sensors were also present.

Furthermore, as specified in line 439, the full description of the phytotron chamber is reported by reference no 31 and 85.

4. We have changed it as suggested.

5. We have described the RWC data in the results section (line 138).